# Coverage and Drivers of Vaccinations in Patients with Autoimmune Rheumatic Diseases: An Italian Multicentric Study

**DOI:** 10.3390/vaccines13121229

**Published:** 2025-12-06

**Authors:** Ilaria Anna Bellofatto, Valentino Paci, Fabrizio Conti, Gianluca Santoboni, Gian Domenico Sebastiani, Maria Sofia Cattaruzza, Camilla Mazzanti, Simonetta Salemi, Giorgio Sesti, Emanuele Tesoriere, Valerio Fiorilli, Imma Prevete, Francesca Romana Spinelli, Andrea Picchianti Diamanti

**Affiliations:** 1Critical Care Medicine Unit, Emergency Department, Ospedali Galliera, 16128 Genova, Italy; ilaria.bellofatto@live.it; 2Medical Clinic, Department of Clinical and Molecular Sciences, Marche Polytechnic University & Department of Internal Medicine, Azienda Ospedaliero-Universitaria delle Marche, 60126 Ancona, Italy; valentinopaci@gmail.com; 3Arthritis Center, Reumatologia, Dipartimento di Scienze Cliniche Internistiche, Anestesiologiche e Cardiovascolari–Sapienza University of Rome, 00185 Roma, Italyvalerio.fiorilli@uniroma1.it (V.F.); francescaromana.spinelli@uniroma1.it (F.R.S.); 4Center for the Diagnosis and Therapy of Autoimmune Rheumatological Diseases, Belcolle Hospital, 01100 Viterbo, Italy; gianluca.santoboni@gmail.com (G.S.); mazzanti.1651413@studenti.uniroma1.it (C.M.); 5U.O.C. Reumatologia, A.O. San Camillo-Forlanini, 00152 Roma, Italyimmaprevete@virgilio.it (I.P.); 6Department of Public Health and Infectious Diseases, Sapienza University of Rome, 00185 Rome, Italy; 7Department of Clinical and Molecular Medicine, S. Andrea University Hospital, “Sapienza” University of Rome, 00185 Rome, Italy; simonettasalemi@gmail.com (S.S.); giorgio.sesti@uniroma1.it (G.S.); tesoriere.1884799@studenti.uniroma1.it (E.T.)

**Keywords:** vaccinations, influenza, pneumococcus, varicella zoster virus, papillomavirus, hepatitis B virus, systemic lupus erythematosus, rheumatoid arthritis, biologic DMARDs, JAK inhibitors

## Abstract

Background: Patients with autoimmune rheumatic diseases (ARDs) such as systemic lupus erythematosus (SLE) and rheumatoid arthritis (RA) are at increased risk of infections due to immune dysregulation and immunosuppressive therapy. Vaccination is a cornerstone of infection prevention, but uptake is still inadequate. Methods: We conducted an observational, multicenter study at four Italian rheumatology centers. Adult patients with RA or SLE on immunosuppressive therapy completed a standardized questionnaire assessing demographics, disease activity, treatments, vaccination status for influenza, pneumococcus, varicella-zoster virus [VZV], hepatitis B virus [HBV], human papillomavirus [HPV], adverse events, and reasons for or against vaccination. Results: A total of 325 patients were included (226 RA, 99 SLE; median age 60 years; 84.6% females). Overall vaccine coverage was 68.0%, with influenza being the most frequent (54.2%), followed by pneumococcal (30.8%), HBV (21.2%), VZV (12.9%) and HPV (5.9%). RA patients showed higher influenza and pneumococcal uptake, while HBV vaccine was more common in SLE. Education was associated with higher pneumococcal and HBV coverage in both groups. Major adverse events and disease flares were rare. Physician recommendation was the main motivator, with rheumatologists driving VZV uptake and general practitioners influencing influenza and HBV. Among unvaccinated patients, the leading barrier was not being offered vaccination (42.5%), followed by concerns about efficacy/safety (18.3%) and lack of awareness (15.7%). Cluster analysis identified three subgroups: “Not offered” (largest), “Unaware,” and “Skeptical,” with age distribution differing across clusters. Conclusions: Vaccination coverage among Italian ARD patients remains insufficient. Physician recommendation is pivotal, while lack of physician offer and awareness are major barriers. Targeted educational strategies and proactive physician involvement are needed to improve vaccine uptake in this high-risk population.

## 1. Introduction

Patients with autoimmune rheumatic diseases (ARDs) such as Systemic Lupus Erythematosus (SLE) and Rheumatoid Arthritis (RA) bear a higher burden of infections, resulting in increased morbidity and mortality compared to the general population. This vulnerability stems from the confluence of inherent immune dysregulation, disease activity, end-organ damage, comorbidities and the immunosuppression needed to achieve treat-to-target goals [1,2,3]. Beyond their immediate morbidity, infections often precipitate disease flares amplifying cumulative harm [4].

As in SLE, patients with RA represent a particularly vulnerable subgroup within ARD population, with a high susceptibility to both common and opportunistic infections, which contribute to excess morbidity and mortality. RA is often accompanied by comorbidities such as interstitial lung disease and cardiovascular disease, that increase the likelihood of complications even further. Taken together, these shared vulnerabilities support the necessity of timely and comprehensive vaccination strategies [1,2].

Vaccines are the most effective strategy to enhance or restore protective immunity, thereby reducing the risks of infection, hospitalization, and mortality. According to WHO estimates, approximately 154 million lives, primarily of children, have been saved through vaccination over the past 50 years [5]. Last recommendations from the American College of Rheumatology (ACR) and the Italian Society of Rheumatology (SIR) [6,7] strongly advise that adults with ARDs on immunosuppressive therapy receive vaccinations against influenza, pneumococcus and varicella-zoster virus (VZV), whereas other vaccines such as human papillomavirus (HPV) and hepatitis B virus (HBV) are suggested or advised as per general population guidelines [6,7]. The safety and efficacy of vaccinations in patients with SLE and RA have been well documented for influenza, pneumococcus, and SARS-CoV-2, while data for VZV (particularly with the recombinant adjuvanted Shingrix^®^ vaccine) and HPV, though generally reassuring, remain more limited [8,9,10,11,12]. Despite these findings, vaccination coverage is still suboptimal. This can be attributed to both general and disease-specific concerns. Vaccine hesitancy is driven by limited health literacy, distrust of medical professionals, and fears of vaccine safety issues. Among patients with ARDs, additional barriers include concerns about triggering disease flares, uncertainty about vaccine efficacy under immunosuppression, and confusion about the appropriate management of immunosuppressive therapy during vaccination cycles [13,14].

To date, there is still a limited number of studies analyzing vaccination coverage in patients with ARD, including SLE and RA [15].

This study aims to quantify vaccination coverage for influenza, pneumococcus, VZV, HBV, and HPV among adults with SLE and RA and to identify factors associated with vaccine uptake or refusal.

## 2. Materials and Methods

This is an observational, cross-sectional, multicentric study including patients with RA and SLE follow-up at the rheumatology outpatient clinic of Sant’Andrea University Hospital in Rome, Policlinico Umberto I in Rome, San Camillo Forlanini in Rome and Belcolle Hospital in Viterbo seen between June 2024 and January 2025. We included adult patients diagnosed with RA or SLE according to EULAR 2010 criteria for RA and EULAR 2019 criteria for SLE. Additionally, patients had to be undergoing immunosuppressive therapy. Data were collected on the day of the visit using a self-administered questionnaire provided to patients by the attending research staff at the beginning of the visit, and participants were informed that their responses were completely voluntary and anonymous, and that declining to participate would have no effect on their care. Participants completed the questionnaire privately in the waiting area or in a separate room, and questionnaires were collected immediately after completion by the research staff. The questionnaire included closed-ended questions, covering the following information: demographic information [age, sex, educational level, smoking habits, body mass index (BMI)], current diagnosis (RA or SLE), comorbidities, and disease activity, assessed using DAS28 score for RA and SLEDAI score for SLE. Ongoing treatment was also recorded, including the use of conventional synthetic Disease Modifying Anti-Rheumatic Drugs (csDMARDs), biological DMARDs (bDMARDs) and targeted synthetic DMARDs (tsDMARDs) along with their respective dosages. We collected data on all inactivated vaccines recommended in Italy, excluding those restricted to pediatric age, while also including vaccines recommended only for specific age-defined subgroups. In this framework, HPV vaccination was recorded despite being offered primarily to adolescents within national catch-up programs. Patients were also asked whether they had recently contracted an infection related to any of these vaccine-preventable diseases. Patients were classified as “vaccinated” if they had received at least one of the specified vaccines. For vaccinated patients, additional information was asked, including the type of vaccine, number of injections, suspension of immunosuppressive therapy, adverse effects and disease flares following vaccination. The questionnaire also investigated the factors influencing the decision to vaccinate, with options such as general practitioner (GP) advice, rheumatology specialist advice, or personal choice. For unvaccinated patients, reasons for non-vaccination were collected as well, with potential reasons including: GP or rheumatology specialist advice against it, lack of knowledge about the vaccine, uncertainty regarding its efficacy or side effects, fear of disease reactivation, lack of advice from a healthcare provider, exceeding the recommended age limit (only for HPV), previous side effects or allergic reactions to vaccines. Education level was categorized into three groups: primary school completion (level 1), secondary school completion (level 2) and higher education (level 3). Levels of immunosuppression were classified into three categories: “mild” for patients on hydroxychloroquine with or without glucocorticoids at a dose of less that 7.5 mg per day; “moderate” for those on methotrexate, cyclosporine, salazopyrin, leflunomide with or without glucocorticoids at a dose between 7.5 and 15 mg per day; “high” for patients receiving glucocorticoids higher than 15 mg/daily, cyclophosphamide, mycophenolate mofetil, any biologic or targeted synthetic DMARDs with or without glucocorticoids. Disease activity for RA was categorized according to the EULAR definitions [16] remission (DAS28 score < 2.6), low (DAS28 score between 2.6 and 3.1), moderate (DAS28 score between 3.2 and 5.1) and high (DAS28 score > 5.1). As for SLE, disease activity was classified into two levels based on the SLEDAI 2K score: Low Disease Activity (SLEDAI 2K < 6) and moderate/high disease activity (SLEDAI 2K > 6) [17]. To harmonize the classifications, we have then combined the remission and low disease activity categories for RA, in order to have two levels of activity for both diseases. Ethical Approval: The study was conducted in accordance with the principles of the Declaration of Helsinki and approved by the ethics committee of the coordinating center (Lazio area 1, Approval numbers 0942/2024 on 25 October 2024) and by each center participating in the study; written informed consent was obtained from all the patients.

### Statistical Analysis

All statistical analyses were performed using R software (version 4.4.2) and a *p*-value less than 0.05 was considered significant. For descriptive analyses, continuous variables were expressed as median with interquartile range (IQR) and categorical variables as absolute numbers and frequencies. Between-group comparisons were performed using the Wilcoxon rank sum test for continuous variables and Pearson’s χ^2^ test or Fisher’s exact test for categorical variables, as appropriate. Vaccine coverage was expressed as proportions with corresponding counts, overall and stratified by diagnosis, age group, gender, education level, and immunosuppression. Differences between subgroups were assessed using Pearson’s χ^2^ or Fisher’s exact test for categorical variables, as appropriate. Continuous variables were compared using the Wilcoxon rank-sum test for two-group comparisons and the Kruskal–Wallis test for multi-group comparisons. To identify independent predictors of overall vaccine coverage, a multivariable logistic regression model was performed, including baseline covariates, and results were reported as odds ratios (OR) with 95% confidence intervals (CI). Reasons in favor of or against vaccination were analyzed descriptively, and a cluster analysis was conducted among unvaccinated patients. All patients who reported at least one reason for non-vaccination were included in the analysis. For each patient, the weighted proportion of each reported reason was computed. Data regarding motivations against HPV vaccination were excluded from this analysis, as eligibility for HPV vaccination in Italy is restricted to specific age groups. Their inclusion would have introduced structural variability and reduced model stability. Distances between patients were calculated using Euclidean metrics, and clustering was performed with Ward’s hierarchical method. Clusters were then compared across demographic and clinical covariates using χ^2^ or Kruskal–Wallis tests, as appropriate. Post hoc examination of standardized residuals from χ^2^ analyses was computed to identify which age groups were over- or under-represented within clusters.

## 3. Results

The study population included 325 patients: 226 (69.5%) with RA and 99 (30.5%) with SLE. Among the more relevant baseline data, females were the majority (84.6%), with a significantly higher proportion in the SLE group (92.9% vs. 81%, *p *= 0.006). Only 21% of the study population had a university degree or higher. The distribution of education level differed significantly between RA and SLE groups: 43.8% of RA patients reported only primary education, while 32.7% of SLE patients had completed tertiary education (*p* < 0.001). Most patients were in remission or had low disease activity (79.5%) (Table 1).

Overall vaccine coverage was 68.0% (221/325), with a similar proportion in the two groups. Influenza vaccination was the most common (54.2%), followed by pneumococcal (30.8%) and HBV (21.2%). VZV and HPV uptake were lower (12.9% and 5.9%, respectively). Influenza and pneumococcal vaccination rates were higher in RA patients (60.2% and 36.3%) compared to SLE (18.2% and 40.4%) (*p* = 0.002 for both); conversely, HBV vaccination was more frequent in SLE (33.3% vs. 15.9%, *p* < 0.001) (Figure 1).

Regarding safety, major adverse events (AEs) were uncommon (1.9% overall), with similar frequencies across groups. Minor AEs occurred in 12.6% of patients (11.5% RA, 15.1% SLE). Disease flares after vaccination were reported by 1.9% of the population, with a not significantly higher rate of patients experiencing flares among SLE patients (4.0% in SLE vs. 0.9% in RA patients).

The analysis of vaccine coverage by covariates revealed that in both RA and SLE, influenza and pneumococcal coverage increased with age, with the highest uptake in RA patients aged ≥65 years (*p* < 0.001). In contrast, HBV and HPV vaccination were predominantly reported among individuals aged 18–49 in both disease groups (*p* < 0.001). RA patients with higher education were more likely to receive pneumococcal and HBV (*p* = 0.024 and *p* = 0.005). Likewise, SLE patients in the higher education group had higher HBV vaccination rates (*p* = 0.011). Neither gender nor level of immunosuppression significantly influenced compliance with any vaccine in either disease group (Table 2).

We then explored possible associations between the main demographic and clinical variables and vaccine compliance among patients in a multivariate analysis. None of the variables considered resulted in a significant independent predictor of vaccination uptake (Appendix A).

### Motivation Analysis

Among vaccinated individuals, the leading reason for vaccination was physician recommendation, either from the GP (41.1%) or the rheumatologist (40.9%). The influence of the recommending physician varied by vaccine: a rheumatology specialist's advice was the main driver for VZV vaccination (71.4%), pneumococcal (50.0%) and HPV vaccination (36.8%), while GPs mostly motivated influenza (47.7%) and HBV vaccination (43.5%). Personal decision played a smaller role in vaccination compliance (12.8%), although it was more frequent for HPV (31.6%). Finally, childhood vaccination accounted for nearly one-third of cases of HBV vaccination (30.4%), in addition to GP advice. Among unvaccinated individuals, the most common reason against vaccines was not being offered by a physician (42.5%). Other frequent barriers included concerns about efficacy and/or adverse events (18.3%), followed by lack of awareness (15.7%), with the latter being particularly prominent for VZV (32.9%). Other motivations less frequently reported were fear of disease flares (2.2%), having contraindications (2.1%), and history of hypersensitivity (2.2%). For HPV specifically, the main reason for non-vaccination was being outside the recommended age group (67.3%) (Figure 2)

Bars represent vaccine-specific proportions, while the line indicates overall distribution across categories. VZV: varicella-zoster virus (vaccine); HPV: human papillomavirus (vaccine); PNV: pneumococcal vaccine; FLU: influenza virus (vaccine); HBV: hepatitis B virus (vaccine).

A cluster analysis was conducted to identify distinct subgroups within the unvaccinated patients, based on their reported reasons against vaccination. Three motivational clusters were identified: “Not offered”, “Unaware” and “Skeptical”. The largest cluster was the “Not offered” (N = 226), followed by the “Skeptical” (N = 47) and the “Unaware” (N = 42). The first cluster was predominantly characterized by the report that vaccination had not been offered by a physician (71.6% of answers), the second one mainly reflected lack of awareness of vaccine availability or recommendation (80.0% of answers), while the third cluster was dominated by concerns about vaccine efficacy and adverse events (88.8% of answers) (Table 3 and Figure 3).

Comparisons across demographic and clinical variables excluded any association between a cluster membership and diagnosis, gender, BMI, educational level or immunosuppression group. However, age distribution differed significantly between clusters (*p = 0.017*). Specifically, post hoc analysis of standardized residuals indicated that patients aged 50–64 were over-represented in the “Unaware” cluster and under-represented in the “Not offered” cluster. Conversely, patients aged 65 or above were over-represented in the “Not offered” and under-represented in the “Unaware” clusters. Finally, no specific age group predominated in the “Skeptical” cluster (Appendix A).

## 4. Discussion

The findings of the study indicate that uptake of influenza, pneumococcus, VZV, HPV and HBV vaccines in patients with RA and SLE is suboptimal, highlighting missed opportunities for prevention. Overall rates align closely with those observed in high-risk groups within the general Italian population, but with notable variations across vaccine types and age groups.

### 4.1. National and International Context

The “Calendario Vaccinale per la Vita”, developed by the Italian Society of Hygiene, Preventive Medicine and Public Health (SITI), clearly outlines the types and timing of vaccinations recommended for children and adults in Italy [18].

The Italian Istituto Superiore di Sanità sets influenza vaccination targets at 75% for minimum and 95% for optimal coverage among the target population, comprising individuals aged 65 and older, as well as high-risk individuals of any age [19]. However, data for the 2024/2025 influenza season, updated as of 29 August 2025, reveal that only 52.5% of people aged 65 or older received the influenza vaccine. The steady decline in influenza vaccination since its peak in the 2020–2021 season (65.3%) suggests the higher coverage achieved during the pandemic has waned, creating challenges in sustaining high vaccination rates among vulnerable groups [19]. In our cohort, uptake was comparable to that of the elderly Italian population (54.2% vs. 52.5%) but remained far below the recommended target, given that all our patients are classified as “at risk”. When focusing specifically on patients aged 65 or older in our cohort, coverage increased to 77.8%, indicating a better compliance once patients enter the age group targeted by national campaigns.

National available data on pneumococcal vaccination primarily refer to the pediatric population. In Italy, the pneumococcal vaccine is administered to infants after the fifth month of life, in combination with a six-valent vaccine that protects against diphtheria, tetanus, pertussis, poliomyelitis, Haemophilus influenzae type B and hepatitis B [20]. No recent national data for pneumococcal vaccination among adults in Italy could be identified. Previous estimates extrapolated from regional registries between 2004 and 2008 reported a maximum coverage of 50% in the adult population [20].

According to WHO statistics, in 2024 pneumococcal vaccination coverage in the general population in Italy was 90.05%, a 1.57% decrease compared with the previous year. However, these data are derived from the WHO/UNICEF Joint Reporting Form on Immunization and mainly reflect the pediatric population [21]. Consequently, there is a substantial discrepancy between these results and vaccination coverage in our cohort, which was much lower at 30.8%.

A recent Italian epidemiological survey involving 2.357 adults reported that only 33.7% of elderly (≥65 years old) had received the pneumococcal vaccination, far below the national vaccination plan’s target of 75% for this age group [19]. Considering the same age group in our population, rates are slightly higher at 51.3%, but still below the recommended level. Our analyses further revealed higher influenza and pneumococcal vaccination rates in RA patients (60.2% and 36.3%) than in those with SLE (40.4% and 18.2%), likely reflecting the older age of RA patients and consequently their greater inclusion in national vaccination campaigns.

In countries with healthcare systems comparable to the Italian one, rates of influenza vaccination uptake among patients with ARDs range from 45% to 80% [13,14]. Consistent with our findings, older adults tend to exhibit the highest compliance. Our data fall at the lower end of these ranges, highlighting systemic weakness in adherence to vaccination strategies in Italy. For pneumococcal vaccination, reported uptake rates vary even more, ranging from 65% to 80% among individuals aged 65 and older [22,23,24,25].

No official national data on VZV vaccination coverage in Italy are currently available. Nonetheless, the national vaccination plan sets a minimum target coverage of 50% for individuals aged 65 or older and for those aged 18 and above deemed to be “at risk” [20]. A recent survey-based epidemiological study involving 10.000 individuals found that only 9.6% of those considered “at risk” had received the recombinant HZ vaccine [26]. Another Italian retrospective observational study conducted during the 2023 active vaccination campaign, reported an uptake of 13.5% among individuals born in 1958, including both recombinant and live attenuated vaccines [27]. Our data were consistent with these surveys, with a 12.9% uptake, and no significant difference in coverage among the two disease groups.

Studies on HZ vaccination in ARD patients typically show lower uptake, particularly in studies conducted when the live attenuated vaccine was the only available, reflecting concerns regarding viral reactivation [25,28,29,30]. Limited information is currently available on HZ vaccination coverage with Shingrix, since HZ subunit vaccine has only been approved in the US in 2017 and in the EU in 2018. A large retrospective analysis conducted among 548,426 American patients with different ARDs indicated a 9.5% coverage with the recombinant zoster vaccine [28]. Similarly, Krasselt et al. reported a coverage of 13.1% in a cohort of 222 adult ARD patients, whereas more encouraging results have been showed in a retrospective observational cohort study conducted among 132,672 American patients with inflammatory arthritis showing a 21.6% uptake with the recombinant vaccine [31,32].

HPV vaccination is offered to all children during their 12th year of life through an active call system in all Italian regions. Many regions also extend eligibility to other age groups. The currently used vaccine is the nine-valent formulation, administered in two doses six months apart; however, individuals aged 15 or above are advised to have three doses [19]. The most recent official Italian data on HPV vaccination rates date back to 2023 and report coverage among age groups included in the active call plan.

By the end of that year, 60.8% of 13-year-old girls had been vaccinated, compared to 51.6% of boys in the same age group. Among girls turning 15 in 2023, who should have completed the entire vaccination cycle, coverage rate was 69.57%, consistent with the previous year [33]. These rates remain far below the optimal target of 95% coverage for both sexes set by the National Prevention Plan [19].

In our cohort, only 5.9% of patients had received the HPV vaccine, although 67.3% of the population exceeded the recommended age limit for vaccination; thus, the low HPV coverage observed in our cohort should be interpreted with caution. Since its introduction, the HPV vaccine has been provided free of charge only to teenage females, and it was only recently made available to both sexes [19]. This may partly explain the lower vaccination coverage seen in older individuals. Few international studies in patients with systemic autoimmune diseases report variable uptake, ranging between 4.6% and 20.6% [34,35,36]. HPV vaccination is particularly influential for patients with SLE, who demonstrate significantly higher rates of HPV infection and HPV-related dysplastic lesions compared with healthy controls. Reported infection rates range from 20.2 to 80.7% in SLE patients compared with 7.3–35.7% in control populations [37,38]. These data reinforce the need to actively promote HPV vaccination among young SLE patients, ideally before the onset of immunosuppressive therapy, to reduce the risk of persistent infection and related complications.

Vaccination for HBV has been mandatory for infants in Italy since 1991, making it the first country in Europe to achieve the WHO regional target of 95% coverage with three doses. Official data on adult HBV vaccination coverage in Italy are lacking, though estimates can be drawn from research studies [20]. An Italian study on 731 patients with liver cirrhosis unrelated to HBV found an overall vaccine coverage of 16.3%, with higher uptake in people under 65 years (23.7%) compared to older individuals (10%) [39]. In our cohort, overall coverage was higher at 21.2% with significantly greater uptake among SLE patients compared to RA patients. This likely reflects that many SLE patients were born after 1991, when HBV vaccination became mandatory. Accordingly, vaccination was significantly more common in the 18–49 age group for both SLE (59.1%) and RA patients (51.5%) and, in particular, in the subgroup of subjects born after 1991 (i.e., aged 18–34 years, 88.9%).

Results from international comparisons demonstrate substantial variation across studies. In a Canadian study, the coverage of HBV vaccination in patients with ARD ranged from 33.6% to 55.6% [40]. By contrast, other studies have reported markedly lower rates, with as few as 11% of patients vaccinated [41].

### 4.2. Drivers of Vaccine Uptake and Barriers

Interestingly, in our study we detected an association between higher level of education and HBV vaccination uptake. This finding aligns with broader evidence suggesting that better health literacy predicts higher uptake of HBV and other vaccines, even among patients with ARDs [42,43]. Multivariate analysis of demographic and clinical variables revealed that none were significant independent predictors of vaccine uptake. In analyzing the reasons behind vaccination decisions, our results confirm the central role of physician recommendation in driving vaccination among patients with RA and SLE. This finding is consistent with reports from other ARD cohorts, where physician advice is consistently valued as the strongest facilitator of vaccine acceptance, while vaccine hesitancy largely comes from concerns regarding efficacy and safety [13,14,22,38,44,45,46,47,48,49,50]. The relative influence of the GP or the rheumatologist varied across vaccines: rheumatologists were the main promoters of VZV, pneumococcal, and HPV vaccination, whereas GPs most frequently encouraged influenza and HBV vaccination. The reliance on physicians likely reflects the complexity of vaccine decision-making in patients with autoimmune diseases taking immunosuppressants. These findings suggest that vaccination patterns are determined by the involvement of different healthcare providers, underlining the need for stronger coordination between primary and specialty care to maximize uptake across vaccine types. As conceivable, among unvaccinated individuals, the most frequent reason was that vaccination had not been offered by a physician. Our cluster analysis further clarified the heterogeneity of unvaccinated patients. The predominance of the “Not offered” cluster reinforces the importance of physicians’ role, given that without a direct recommendation, most patients remain unvaccinated. Age distribution across clusters differed significantly and is particularly informative. Patients aged 50–64 years were over-represented in the “Unaware” group and under-represented in the “Not offered” group, highlighting the fact that this transitional age group is at risk of being overlooked by age-based recommendations. Patients aged 65 or above showed the opposite pattern, suggesting that despite established age-based recommendations, missed opportunities in primary and hospital care persist. Other reported barriers were concerns about efficacy, accounting for the “Skeptical” cluster, and/or adverse events and lack of awareness, the latter being especially relevant for VZV and underscoring the need for stronger dissemination of information about the recombinant zoster vaccine. By contrast, fears of disease flares, contraindications, or hypersensitivity reactions were rarely cited, suggesting that vaccine hesitancy in this group is not primarily driven by concerns related to the underlying disease.

### 4.3. Limitations

A main limit of the present study is represented by the relatively small size of the sample, which does not allow for generalizing results to the entire RA and SLE population in Italy. Moreover, our cohort was recruited from a single region; therefore, regional differences in vaccination programs and physician practices across Italy may not be fully represented. In addition, vaccination status was self-reported, which may cause recall bias, especially for vaccines received many years earlier.

## 5. Conclusions

Our findings highlight substantial gaps in vaccination coverage among RA and SLE patients in Italy, with important variations by vaccine type, age group, and disease category. Physicians’ recommendation strongly influences uptake, while lack of offers and awareness remain major barriers. Improving coverage will require coordinated efforts between primary and specialty care, targeted education, and systematic vaccination strategies. Addressing these gaps is critical to reducing preventable infections in high-risk autoimmune disease populations.

## Figures and Tables

**Figure 1 vaccines-13-01229-f001:**
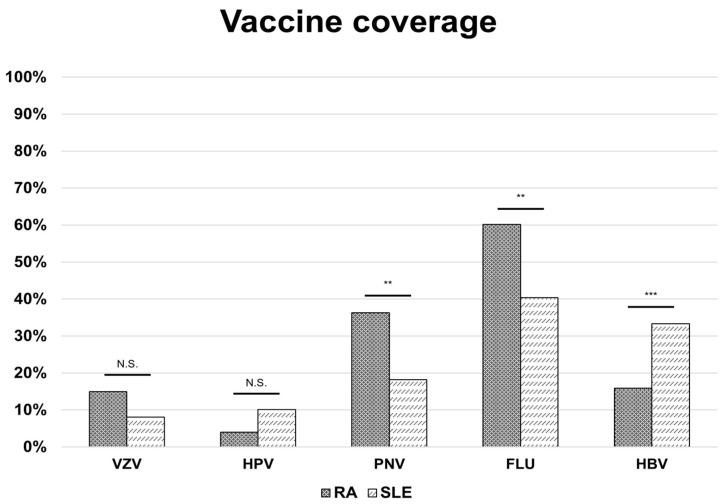
Vaccine coverage in patients with rheumatoid arthritis and systemic lupus erythematosus. Bars represent disease-specific proportions, while the red line indicates the overall vaccination rate across groups. *p*-values significant if <0.05; ** = *p*-value < 0.01; *** = *p*-value < 0.001; N.S. = not significant. Abbreviations: RA: rheumatoid arthritis; SLE: systemic lupus erythematosus; VZV: varicella-zoster virus (vaccine); HPV: human papillomavirus (vaccine); PNV: pneumococcal vaccine; FLU: influenza virus (vaccine); HBV: hepatitis B virus (vaccine).

**Figure 2 vaccines-13-01229-f002:**
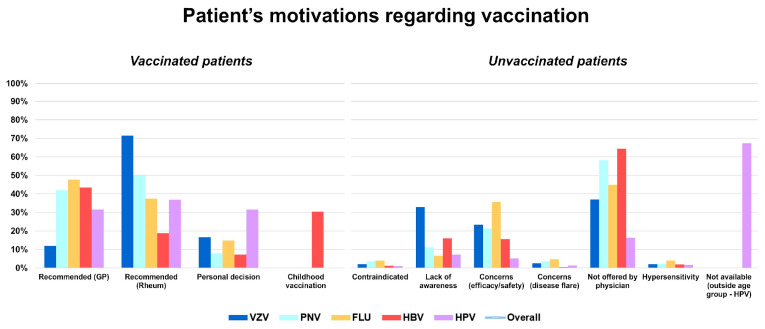
Patient-reported motivations for receiving (left graph) or not receiving (right graph) vaccination.

**Figure 3 vaccines-13-01229-f003:**
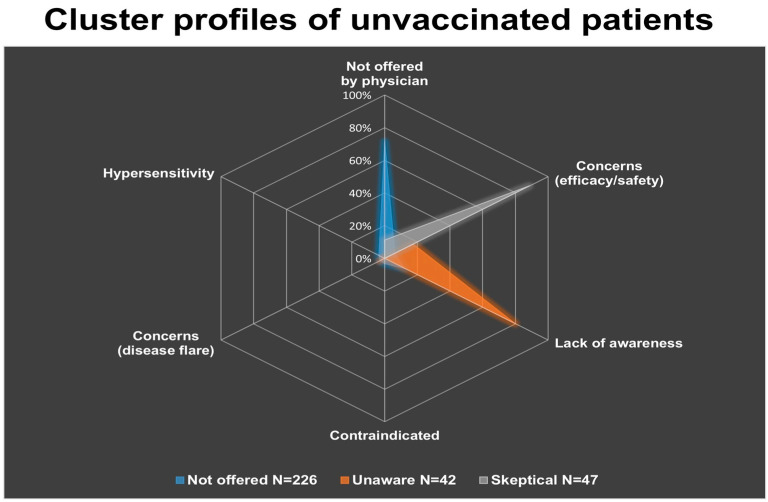
Cluster profiles of unvaccinated patients, derived from the weighted distribution of reported reasons against vaccination (excluding HPV). Three distinct groups emerged: the “Not offered” cluster (N = 226), predominantly characterized by lack of physician recommendation (71.6%); the “Unaware” cluster (N = 42), mainly driven by lack of awareness (80.0%); and the “Skeptical” cluster (N = 47), primarily associated with concerns about efficacy and/or adverse events (88.8%). Other factors, such as physician contraindication, fear of disease flare, or history of hypersensitivity, contributed minimally within clusters.

**Table 1 vaccines-13-01229-t001:** Baseline demographic and clinical characteristics of the entire cohort and stratified by diagnosis.

	Overall N = 325	RA N = 226	SLE N = 99	*p*-Value
Gender, (%)				0.006
Females	275 (84.6%)	183 (81.0%)	92 (92.9%)	
Males	50 (15.4%)	43 (19.0%)	7 (7.1%)	
Age, yrs, median [IQR]	60.0 [50.0–70.0]	64.0 [55.0–74.0]	51.0 [41.0–59.0]	<0.001
Group 18–49, (%)	77 (23.7%)	33 (14.6%)	44 (44.4%)	
Group 50–64, (%)	131 (40.3%)	84 (37.2%)	47 (47.5%)	
Group ≥ 65, (%)	117 (36.0%)	109 (48.2%)	8 (8.1%)	
BMI, median [IQR]	24.2 [22.4–27.2]	24.3 [22.5–27.2]	24.1 [21.7–27.4]	0.376
Smoking status, (%)				0.985
Never	209 (64.3%)	146 (64.6%)	63 (63.6%)	
Previous	60 (18.5%)	41 (18.1%)	19 (19.2%)	
Current	56 (17.2%)	39 (17.3%)	17 (17.2%)	
Education level, (%)				<0.001
Primary school	120 (37.0%)	99 (43.8%)	21 (21.4%)	
High school diploma	136 (42.0%)	91 (40.3%)	45 (45.9%)	
University degree	68 (21.0%)	36 (15.9%)	32 (32.7%)	
*Unknown*	1	0	1	
Disease Activity ^a^, (%)				n.a.
Remission/Low	252 (79.5%)	174 (78.4%)	78 (82.1%)	
Moderate/High	65 (20.5%)	48 (21.6%)	17 (17.9%)	
*Unknown*	8	4	4	
Comorbidities, (%)				0.224
Absent	137 (42.2%)	90 (39.8%)	47 (47.5%)	
Present	188 (57.8%)	136 (60.2%)	52 (52.5%)	
Immunosuppression level, (%)				<0.001
Mild	68 (21.1%)	17 (7.6%)	51 (52.0%)	
Moderate	36 (11.2%)	23 (10.3%)	13 (13.3%)	
High	218 (67.7%)	184 (82.1%)	34 (34.7%)	
*Unknown*	3	2	1	

*p*-values refer to comparisons between RA and SLE (Wilcoxon rank-sum for continuous, χ^2^ for categorical variables). a. Disease activity was assessed using DAS-28 and SLEDAI scores for patients affected by RA and SLE, respectively. Patients were considered in remission/low disease activity with a DAS-28 ≤ 3.2 or SLEDAI < 6. Abbreviations: RA, rheumatoid arthritis; SLE, systemic lupus erythematosus; BMI, body mass index; DAS28, Disease Activity Score on 28 joints; SLEDAI, Systemic Lupus Erythematosus Disease Activity Index, IQR, interquartile range.

**Table 2 vaccines-13-01229-t002:** Vaccine coverage by baseline covariates in patients with rheumatoid arthritis (RA) and systemic lupus erythematosus (SLE). Data are reported as the number (proportion) of vaccinated patients in each subgroup. *p*-values refer to χ^2^ test (or Fisher’s exact test as appropriate) for comparisons within subgroups.

Vaccine by Gender	RA				SLE			
	Males N = 43	Females N = 183		*p* Value	Males N = 7		Females N = 92	*p* Value
VZV, (%)	8 (18.6%)	26 (14.2%)		0.625	2 (28.6%)		6 (6.5%)	0.098
HPV, (%)	-	9 (4.9%)		0.213	-		10 (10.9%)	>0.999
PNV, (%)	16 (37.2%)	66 (36.1%)		>0.999	1 (14.3%)		17 (18.5%)	>0.999
FLU, (%)	27 (62.8%)	109 (59.6%)		0.829	2 (28.6%)		38 (41.3%)	0.698
HBV, (%)	4 (9.3%)	32 (17.5%)		0.249	4 (57.1%)		29 (31.5%)	0.217
Vaccine by age group	Age 18–49 N = 33	Age 50–64 N = 84	Age ≥ 65 N = 109	*p *-value	Age 18–49 N = 44	Age 50–64 N = 47	Age ≥ 65 N = 8	*p *-value
VZV, (%)	6 (18.2%)	10 (11.9%)	18 (16.5%)	0.581	2 (4.5%)	5 (10.6%)	1 (12.5%)	0.426
HPV, (%)	8 (24.2%)	1 (1.2%)	-	*<0.001*	9 (20.5%)	1 (2.1%)	-	*0.011*
PNV, (%)	5 (15.2%)	21 (25.0%)	56 (51.4%)	*<0.001*	6 (13.6%)	8 (17.0%)	4 (50.0%)	0.071
FLU, (%)	10 (30.3%)	41 (48.8%)	85 (78.0%)	*<0.001*	10 (22.7%)	24 (51.1%)	6 (75.0%)	*0.001*
HBV, (%)	17 (51.5%)	8 (9.5%)	11 (10.1%)	*<0.001*	26 (59.1%)	6 (12.8%)	1 (12.5%)	*<0.001*
Vaccine by education level	Primary school N = 99	High school N = 91	University N = 36	*p *-value	Primary school N = 21	High school N = 45	University N = 32	*p *-value
VZV, (%)	15 (15.2%)	13 (14.3%)	6 (16.7%)	0.944	2 (9.5%)	4 (8.9%)	2 (6.2%)	>0.999
HPV, (%)	2 (2.0%)	5 (5.5%)	2 (5.6%)	0.393	1 (4.8%)	5 (11.1%)	4 (12.5%)	0.758
PNV, (%)	45 (45.5%)	24 (26.4%)	13 (36.1%)	*0.024*	7 (33.3%)	6 (13.3%)	5 (15.6%)	0.131
FLU, (%)	65 (65.7%)	49 (53.8%)	22 (61.1%)	0.250	12 (57.1%)	13 (28.9%)	15 (46.9%)	0.065
HBV, (%)	7 (7.1%)	20 (22.0%)	9 (25.0%)	*0.005*	2 (9.5%)	16 (35.6%)	15 (46.9%)	*0.011*
Vaccine by immunosuppression level	Mild N = 17	Moderate N = 23	High N = 184	*p *-value	Mild N = 51	Moderate N = 13	High N = 34	*p *-value
VZV, (%)	4 (23.5%)	1 (4.3%)	28 (15.2%)	0.257	5 (9.8%)	-	3 (8.8%)	0.773
HPV, (%)	-	-	9 (4.9%)	0.806	5 (9.8%)	-	5 (14.7%)	0.367
PNV, (%)	7 (41.2%)	8 (34.8%)	66 (35.9%)	0.900	9 (17.6%)	-	9 (26.5%)	0.094
FLU, (%)	11 (64.7%)	15 (65.2%)	109 (59.2%)	0.796	23 (45.1%)	5 (38.5%)	12 (35.3%)	0.655
HBV, (%)	4 (23.5%)	3 (13.0%)	28 (15.2%)	0.635	18 (35.3%)	5 (38.5%)	10 (29.4%)	0.791

Abbreviations: RA, rheumatoid arthritis; SLE, systemic lupus erythematosus; VZV, varicella zoster virus (vaccine); HPV, human papillomavirus (vaccine); PNV, pneumococcal vaccine; FLU, flu (vaccine); HBV, hepatitis B virus (vaccine).

**Table 3 vaccines-13-01229-t003:** Baseline demographic and clinical characteristics distribution across motivational clusters in the overall cohort.

	Not Offered N = 226	Unaware N = 42	Skeptical N = 47	*p* Value
Diagnosis				
RA	161 (71.2%)	24 (57.1%)	34 (72.3%)	0.171
SLE	65 (28.8%)	18 (42.9%)	13 (27.7%)	
Gender				
Males	34 (15.0%)	8 (19.0%)	6 (12.8%)	0.704
Females	192 (85.0%)	34 (81.0%)	41 (87.2%)	
Age (years)				
Median age [IQR]	61.0 [49.2–73.0]	56.5 [50.2–64.0]	60.0 [52.5–67.0]	0.224
18–49	57 (25.2%)	9 (21.4%)	9 (19.1%)	0.017
50–64	78 (34.5%)	25 (59.5%)	23 (48.9%)	
≥65	91 (40.3%)	8 (19.0%)	15 (31.9%)	
BMI (kg/m^2^)				
Median BMI [IQR]	24.1 [22.3–27.3]	23.6 [20.9–26.4]	24.9 [23.3–26.7]	0.088
Education level				
Primary school	81 (36.0%)	13 (31.0%)	22 (46.8%)	0.509
High School	97 (43.1%)	18 (42.9%)	18 (38.3%)	
University	47 (20.9%)	11 (26.2%)	7 (14.9%)	
*Unknown*	1	0	0	
Immunosuppression level				
Mild	42 (18.8%)	14 (33.3%)	10 (21.3%)	0.124
Moderate	22 (9.8%)	5 (11.9%)	8 (17.0%)	
High	160 (71.4%)	23 (54.8%)	29 (61.7%)	
*Unknown*	2	0	0	

Patients were classified into three mutually exclusive clusters based on their main reported reasons for not being vaccinated: the “Not offered” (N = 226), “Unaware” (N = 42), and “Skeptical” (N = 47) clusters. All patients who reported at least one reason for non-vaccination were included in the cluster analysis. Data regarding motivations against HPV vaccination were excluded from this analysis because HPV vaccination eligibility in Italy is restricted to specific age groups, making motivational patterns not directly comparable with other vaccines. *p* values represent global comparisons across clusters (χ^2^ test for categorical variables; Kruskal–Wallis test for continuous variables). Abbreviations: RA, rheumatoid arthritis; SLE, systemic lupus erythematosus; BMI, body mass index; IQR, interquartile range.

## Data Availability

The original contributions presented in this study are included in the article/Appendix A. Further inquiries can be directed to the corresponding author.

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
