# Peer review of "Coverage and Drivers of Vaccinations in Patients with Autoimmune Rheumatic Diseases: An Italian Multicentric Study"

_vaccines, 2025, doi:10.3390/vaccines13121229_

Round 1
Reviewer 1 Report
Comments and Suggestions for Authors
Estimated Authors,
I've read with interest the present study on the coverage and drivers of vaccinations in Patients with Autoimmune Rheumatic Diseases among a total of 226 patients with rheumatoid arthritis and 99 patients with systemic lupus erythematosus from medical centers in Latium region in Central Italy. Authors documented relatively low vaccination rates for assessed vaccines (i.e. VZV, HPV (the lowest ones), PNV, FLU, and even HBV.
From my point of view, the paper is quite and fully interesting, as it could contribute to fill a knowledge gap in this very critical subpopulation.
However, before the eventual acceptance, I would recommend a couple of adjustments to the present version, as follows:
1) Figure 1, Figure 2: remove the overall line as it does not add any information and makes more complicated accurately follow the paper;
2) Table 2: even though the table is perfectly readable, as the rows quietly and repetitively change across the subsections, did you even tried to transpose the table in order to put vaccines as the headlined for ordinates?
3) again Table 2, Authors should discuss in further details some anomalies they did identify. For example, they should address more carefully that HPV has been available at no costs originally in teen-age females, and only more recently also to men, leading to low vaccination rates in older age groups, in following decades (that is now from this P.O.V.)
4) Authors should explain more carefully how participants were recruited. Currently, the paper states that "Data were collected the day of the visit using a self-administered questionnaire provided to patients" but details such as HOW the questionnaire was delivered and HOW was obtained from participants are still lacking. Authors should report this specific point, as participants feeling themselves obliged to participate in a study may be less accurate in reporting perceived barriers and facilitarors.
5) implementing an adapted version of "Calendario Vaccinale per la Vita" from SITI could improve the readability of your paper.
6) I suggest to design section 4 identifying a series of subheadings (e.g. international contest, vaccination strategies, limitations)
Author Response
I've read with interest the present study on the coverage and drivers of vaccinations in Patients with Autoimmune Rheumatic Diseases among a total of 226 patients with rheumatoid arthritis and 99 patients with systemic lupus erythematosus from medical centers in Latium region in Central Italy. Authors documented relatively low vaccination rates for assessed vaccines (i.e. VZV, HPV (the lowest ones), PNV, FLU, and even HBV. From my point of view, the paper is quite and fully interesting, as it could contribute to fill a knowledge gap in this very critical subpopulation. However, before the eventual acceptance, I would recommend a couple of adjustments to the present version, as follows:
- Figure 1, Figure 2: remove the overall line as it does not add any information and makes more complicated accurately follow the paper;
Figures have been modified as requested
- Table 2: even though the table is perfectly readable, as the rows quietly and repetitively change across the subsections, did you even tried to transpose the table in order to put vaccines as the headlined for ordinates?
We thank the reviewer for this thoughtful suggestion. We explored the possibility of transposing Table 2 by placing vaccines as column headers. We found, however, that given the large number of subgroups included in the analysis, the transposed layout resulted in a less accessible table, ultimately reducing clarity for the reader. Considering this, we opted to retain the current structure
- Again Table 2, Authors should discuss in further details some anomalies they did identify. For example, they should address more carefully that HPV has been available at no costs originally in teen-age females, and only more recently also to men, leading to low vaccination rates in older age groups, in following decades (that is now from this P.O.V.)
We appreciate the reviewer’s insightful comment and have incorporated an explanatory note in lines 395–398
4) Authors should explain more carefully how participants were recruited. Currently, the paper states that "Data were collected the day of the visit using a self-administered questionnaire provided to patients" but details such as HOW the questionnaire was delivered and HOW was obtained from participants are still lacking. Authors should report this specific point, as participants feeling themselves obliged to participate in a study may be less accurate in reporting perceived barriers and facilitarors.
We thank the reviewer for this important comment. Detailed data have been added to the text in the methods section
5) implementing an adapted version of "Calendario Vaccinale per la Vita" from SITI could improve the readability of your paper.
We thank the reviewer for this helpful suggestion. We mentioned and cite the SITI calendar in the discussion section.
6) I suggest to design section 4 identifying a series of subheadings (e.g. international contest, vaccination strategies, limitations)
Subheadings nave been added to the discussion
Reviewer 2 Report
Comments and Suggestions for Authors
Dear Authors!
Thank you for the opportunity to read and review your manuscript!
Patients with immune-mediated diseases have higher risk of infections and have risk to have a more severe course due to baseline immune system dysregulation, immunosupressive drugs. Vaccines are the mosrt reliable tool for prevent and control of vaccine-controlled diseases, but the coverage of such patients usualy lower compared to population.
The finding of the barriers for such vaccination and improvement the situation is an actual part of practical work of rheumatologists
In this study Authors provided the cross-sectional study to find the barriers for vaccinations and provided the comprehensive analysis.
Th cluster analysis proviede by the Authors provided three groups
In the discussion Authors cpompared their data with the previously published results. The manuscript has only relevant literature
The discussion has the limitation section
The conclusion supprots the study results
During the revision I have only one optinal suggestion to rpovide the multivariate logistic regression analytsis to find the vaccination barries
Good luck!
Author Response
Thank you for the opportunity to read and review your manuscript!
Patients with immune-mediated diseases have higher risk of infections and have risk to have a more severe course due to baseline immune system dysregulation, immunosupressive drugs. Vaccines are the mosrt reliable tool for prevent and control of vaccine-controlled diseases, but the coverage of such patients usualy lower compared to population. The finding of the barriers for such vaccination and improvement the situation is an actual part of practical work of rheumatologists In this study Authors provided the cross-sectional study to find the barriers for vaccinations and provided the comprehensive analysis. Th cluster analysis proviede by the Authors provided three groups In the discussion Authors cpompared their data with the previously published results. The manuscript has only relevant literature The discussion has the limitation section The conclusion supprots the study results
During the revision I have only one optinal suggestion to rpovide the multivariate logistic regression analytsis to find the vaccination barries
Good luck!
We thank the reviewer for his positive comments and for highlighting the clinical relevance of our study. We truly appreciate the time taken to review our manuscript and the encouraging feedback
Reviewer 3 Report
Comments and Suggestions for Authors
Dear authors
I would like to thank you for the opportunity you have given me to considerate the very interesting article entitled “Coverage and drivers of vaccinations in Patients with Autoimmune Rheumatic Diseases: An Italian multicentric study” by Bellofatto and colleagues were submitted for consideration and probably publication on Vaccines Journal. The manuscript addresses a highly sensitive scientific topic—namely, the vaccination of patients with autoimmune rheumatoid arthritis. The primary concern in such cases, and more broadly among immunosuppressed patients, is maintaining the balance between adequate vaccine coverage and patient safety, particularly when vaccines containing live attenuated strains are involved.
Nevertheless, the authors engage with a timely and important issue regarding the immunization of special patient groups, and they present a well-written manuscript with clear focus and results. The text requires only minor improvements. My decision is “minor revision,” and my suggestions for improving the manuscript follow.
Introduction
Strengthen your introduction with an additional paragraph highlighting the specific category of patients with rheumatoid arthritis and the necessity of vaccination, beyond the paragraph that addresses vaccine safety and efficacy (lines 66–69).
Lines 76-77 “To date, there is still a limited number of studies analysing vaccination coverage in patients with ARD, including SLE and RA”. Suggested to add reference to enhance your position (Facilitators and barriers of vaccine uptake in patients with autoimmune inflammatory rheumatic disease: a scoping review. Neusser S, et al. RMD Open. 2022 Dec;8(2): e002562. doi: 10.1136/rmdopen-2022-002562).
Materials and Methods
Line 86: add the study period
Results
- Suggested to authors to remove from the first column of the table the (n) because this is listed in the first row of the table and it is redundant to also list it in the first column of the table
- Lines 222-223: The authors correctly note in their results that the primary reason for non-vaccination against HPV is the absence of a recommendation for these age groups. This point should also be clarified in the methodology of their study, explaining why this vaccine it was included to measure the vaccination coverage for this specific group. Using it as a comparative indicator would constitute a sound approach.
- In every instance where you state that the vaccine was not offered, please clarify that this refers to it not being offered by physicians. This will help prevent potential confusion among readers who might otherwise assume that the vaccine was not offered by the Italian National Immunization Program.
- Additionally, if data on participants’ place of residence are available, it would be of interest to compare individuals living in urban areas with those residing in non-urban areas.
Discussion
Lines 269-273: please add reference (or national data) about the vaccination coverage on high risk groups in Italy.
Lines 360-364: The authors’ approach regarding individuals born after 1991 and hepatitis B vaccination is very appropriate. You could substantially strengthen your argument by further analysing this age group, as despite exhibiting the highest vaccination coverage at 51.5% for males compared with the other age groups, the rate remains low for a vaccine that was introduced into Italy’s national immunization program in the 1990s. Do the authors have any explanation for this? Perhaps conducting a sub-analysis within the group itself would be beneficial, as the vaccination coverage exhibits a substantial discrepancy between male and female participants.
Author Response
I would like to thank you for the opportunity you have given me to considerate the very interesting article entitled “Coverage and drivers of vaccinations in Patients with Autoimmune Rheumatic Diseases: An Italian multicentric study” by Bellofatto and colleagues were submitted for consideration and probably publication on Vaccines Journal. The manuscript addresses a highly sensitive scientific topic—namely, the vaccination of patients with autoimmune rheumatoid arthritis. The primary concern in such cases, and more broadly among immunosuppressed patients, is maintaining the balance between adequate vaccine coverage and patient safety, particularly when vaccines containing live attenuated strains are involved.
Nevertheless, the authors engage with a timely and important issue regarding the immunization of special patient groups, and they present a well-written manuscript with clear focus and results. The text requires only minor improvements. My decision is “minor revision,” and my suggestions for improving the manuscript follow.
Introduction
Strengthen your introduction with an additional paragraph highlighting the specific category of patients with rheumatoid arthritis and the necessity of vaccination, beyond the paragraph that addresses vaccine safety and efficacy (lines 66–69).
Thank you for this helpful suggestion. We have expanded the introduction by adding a new paragraph that specifically addresses patients with rheumatoid arthritis and outlines the particular reasons vaccination is essential in this population. This new content appears in lines 60–66 of the revised manuscript.
Lines 76-77 “To date, there is still a limited number of studies analysing vaccination coverage in patients with ARD, including SLE and RA”. Suggested to add reference to enhance your position (Facilitators and barriers of vaccine uptake in patients with autoimmune inflammatory rheumatic disease: a scoping review. Neusser S, et al. RMD Open. 2022 Dec;8(2): e002562. doi: 10.1136/rmdopen-2022-002562)
Thank you for this helpful suggestion. We have now added the recommended reference (Neusser et al., RMD Open, 2022) to support our statement regarding the limited number of studies evaluating vaccination coverage in patients with autoimmune rheumatic diseases. The citation has been incorporated at lines 550-552 in the revised manuscript
Materials and Methods
Line 86: add the study period
As requested, the study period has been detailed in the Materials and Methods section.
Results
- Suggested to authors to remove from the first column of the table the (n) because this is listed in the first row of the table and it is redundant to also list it in the first column of the table
We modified the tables as suggested
- Lines 222-223: The authors correctly note in their results that the primary reason for non-vaccination against HPV is the absence of a recommendation for these age groups. This point should also be clarified in the methodology of their study, explaining why this vaccine it was included to measure the vaccination coverage for this specific group. Using it as a comparative indicator would constitute a sound approach.
We thank the reviewer for this valuable observation. Our intention when designing the study was to report coverage for all vaccines that have an official recommendation in Italy (except for those restricted to childhood programmes) and are recommended in patients with autoimmune rheumatic diseases. We agree that this introduces an inherent age-dependent limitation in our cohort, where most participants fall outside the recommended age group. We have clarified this aspect in the Methods section
- In every instance where you state that the vaccine was not offered, please clarify that this refers to it not being offered by physicians. This will help prevent potential confusion among readers who might otherwise assume that the vaccine was not offered by the Italian National Immunization Program.
We thank the reviewer for highlighting this potential source of confusion. Throughout the manuscript, “not offered” have been generally specified as not offered by physicians. We have revised the text to make this point clearer as suggested
- Additionally, if data on participants’ place of residence are available, it would be of interest to compare individuals living in urban areas with those residing in non-urban areas.
This is an interesting point. Unfortunately, we did not register data on patients’ addresses to specifically quantify the proportion of them coming from non-urban areas. However, as far as we are concerned, most of them live in the city of Rome.
Discussion
Lines 269-273: please add reference (or national data) about the vaccination coverage on high risk groups in Italy.
Data have been extrapolated from available observational studies (with the relative reference), because we could not find official national data on vaccination within high-risk groups in Italy, except for influenza vaccination.
Lines 360-364: The authors’ approach regarding individuals born after 1991 and hepatitis B vaccination is very appropriate. You could substantially strengthen your argument by further analysing this age group, as despite exhibiting the highest vaccination coverage at 51.5% for males compared with the other age groups, the rate remains low for a vaccine that was introduced into Italy’s national immunization program in the 1990s. Do the authors have any explanation for this? Perhaps conducting a sub-analysis within the group itself would be beneficial, as the vaccination coverage exhibits a substantial discrepancy between male and female participants.
We thank the reviewer for this insightful comment. We would like to clarify that the 51.5% vaccination rate reported in Table 2 does not refer to male participants, but rather to patients with rheumatoid arthritis aged 18–49 years who received the HBV vaccine. As reported in the manuscript, this age group showed a significantly higher vaccination coverage compared with older age groups. Whereas we did not observe any significant difference in HBV vaccination uptake between male and female patients in our cohort. As suggested by the reviewer, we performed an additional sub-analysis within the 18–49-year age group including both RA and SLE patients, comparing individuals born after 1991 (i.e. aged 18–34 years), who were covered by the mandatory infant vaccination program, with those born before 1991. As expected, HBV vaccination coverage was significantly higher among patients born after 1991. A comment on these data have been added to the Discussion section.
Round 2
Reviewer 1 Report
Comments and Suggestions for Authors
Estimated Authors,
the paper has been amended in accord with my previous remarks.
I've no further requests.